# Packaging of Genomic RNA in Positive-Sense Single-Stranded RNA Viruses: A Complex Story

**DOI:** 10.3390/v11030253

**Published:** 2019-03-13

**Authors:** Mauricio Comas-Garcia

**Affiliations:** 1Research Center for Health Sciences and Biomedicine (CICSaB), Universidad Autónoma de San Luis Potosí (UASLP), Av. Sierra Leona 550 Lomas 2da Seccion, 72810 San Luis Potosi, Mexico; mauricio.comas@uaslp.mx; 2Department of Sciences, Universidad Autónoma de San Luis Potosí (UASLP), Av. Chapultepec 1570, Privadas del Pedregal, 78295 San Luis Potosi, Mexico

**Keywords:** (+)ssRNA viruses, RNA packaging, virion assembly, packaging signals, RNA replication

## Abstract

The packaging of genomic RNA in positive-sense single-stranded RNA viruses is a key part of the viral infectious cycle, yet this step is not fully understood. Unlike double-stranded DNA and RNA viruses, this process is coupled with nucleocapsid assembly. The specificity of RNA packaging depends on multiple factors: (i) one or more packaging signals, (ii) RNA replication, (iii) translation, (iv) viral factories, and (v) the physical properties of the RNA. The relative contribution of each of these factors to packaging specificity is different for every virus. In vitro and in vivo data show that there are different packaging mechanisms that control selective packaging of the genomic RNA during nucleocapsid assembly. The goals of this article are to explain some of the key experiments that support the contribution of these factors to packaging selectivity and to draw a general scenario that could help us move towards a better understanding of this step of the viral infectious cycle.

## 1. Introduction

Nucleocapsid assembly and the RNA replication of positive-sense single-stranded RNA [(+)ssRNA] viruses occur in the cytoplasm. In almost all (+)ssRNA viruses, nucleocapsid assembly requires packaging of the genomic RNA (gRNA) in an infected cell as nucleocapsids cannot be assembled without RNA being packaged. However, there are at least two exceptions to this rule. In vivo assembly of some viruses from the *Picornaviridae* [1,2,3] and *Secoviridae* [4] families results in a small fraction of capsid proteins assembling into empty capsids. Nonetheless, these empty capsids do not contain packaging motors (or similar proteins) and thus cannot package any RNA. From the point of view of assembly, one of the most interesting characteristics of these viruses is that nucleocapsid assembly and RNA packaging are spontaneous processes (no ATP is needed), they are coupled with each other, and depend on RNA-protein and protein-protein interactions [5,6,7,8,9,10,11,12,13,14,15,16,17,18,19,20,21]. This scenario is very different from the assembly of double-stranded DNA and RNA viruses. These viruses follow the “*ds-genome rule*”: first an empty capsid is spontaneously assembled, by protein-protein interactions, and then the genome is packaged into the empty capsid by an enzyme that hydrolyzes ATP (a non-spontaneous process) [22,23,24,25,26,27,28]. There are two viral families that do not follow this rule: *Polyomaviridae* (e.g., simian vacuolating 40 virus) and *Papillomaviridae* (e.g., human papilloma virus). Nucleocapsid assembly in these two families requires the presence of histone-like proteins [29,30] and follows a mechanism that resembles that of (+)ssRNA viruses [31,32,33].

For future reference, it should be noted that in this article we refer to in vitro experiments as those carried out in a test tube and involving purified components and not in cultured cells. The reason for this distinction will become clear later on. In the simplest scenario, in vitro approaches have shown that RNA packaging selectivity depends on highly specific interactions between the gRNA and the capsid protein (CP) [34,35,36,37]. The structural element (and/or sequence) in the gRNA responsible for these interactions is often referred to as a “packaging signal” (PS). As we will discuss in the following sections, PSs are usually associated with the selective packaging of the gRNA. However, there are multiple types and mechanisms by which PSs can contribute to the packaging of the gRNA (see Figure 1).

In vivo and in cultured cells experiments have shown that specific gRNA-CP interactions are necessary but not sufficient to explain packaging specificity observed during an infection. These experiments have demonstrated that there are other factors that contribute to the specificity of this process: (i) the interaction between the RNA-dependent RNA polymerase (RdRp) and gRNA/CP complexes [38], (ii) the length of the packaged RNA [39,40], (iii) the ability of the RNA to be used as a template for RNA replication and/or translation [41,42,43,44], and/or (iv) the compartmentalization of the assembly sites (i.e., viral factories) [45,46,47,48,49,50,51,52].

One of the reasons that makes the study of (+)ssRNA viruses so attractive is that even viruses that are evolutionarily very distant from each other can share a large number of similarities (e.g., genetic, physical, biochemical and/or biological). Therefore, it is too tempting to extrapolate findings from a particular virus to all members of its family, or even to all viruses. However, the specific details of RNA replication, assembly sites, assembly pathways and packaging specificity can differ even between closely related viruses.

The goal of this article is to summarize some key experiments that can help us to understand the interactions that contribute to the ability of (+)ssRNA viruses to selectively package their genomes during an infection (see Table 1). It is important to always keep in mind that the gRNA has to compete against a large and diverse population of cellular and viral-derived (e.g., messenger, subgenomic and/or defective interfering) RNAs. Thus, the virus has to have a strategy to package only the genomic RNA. In this article we compare results from in vitro, in vivo (plants) and cultured cell experiments to draw a general picture that takes into account multiple mechanisms and interactions. It is not our intention to give more importance to one mechanism or another, but to find common ground that can help us to understand RNA packaging. The motivation of this article is to start a discussion about the diversity of mechanisms by which (+)ssRNA viruses package their genome. To do this we will ask the following questions: *do packaging signals exist? To what extent can packaging signals explain specific packaging in an infected cell? How do packaging signals work? What other mechanisms can explain packaging selectivity?*

## 2. General Features of (+)ssRNA Virus Nucleocapsid Assembly

The assembly of nucleocapsids requires, at least, multiple copies of one gene product (usually called capsid or nucleocapsid protein) and one copy of the gRNA. This apparent structural simplicity is true for most plant viruses [53]. The (+)ssRNA bacteriophages are slightly more complex; for example, viruses from the *Leviviridae* family (e.g., MS2) have a second gene product (maturation protein) that is required for in vivo [54] but not in vitro assembly [55,56]. Animal viruses are usually more complex than bacteriophages and plant viruses; many of them incorporate multiple copies of glycoproteins (membrane proteins) and a host-cell derived membrane [57,58]. These glycoproteins identify the appropriate cell receptors and mechanically help the virus to enter the cell [59]. However, members of the *Caliciviridae* [60,61] and *Picornaviridae* [62] families are non-enveloped viruses that do not contain membrane proteins. Furthermore, these viruses, as well as some plant-infecting ones, package another protein (VPg) that is covalently linked to the 5’-end of the gRNA. This protein is required for RNA replication and packaging [53,63]. The role of VPg in RNA packaging will not be discussed in this review.

The above-mentioned structural complexity is not exclusive to the proteinic components of the virion [60]; there are viruses that divide their genome into several molecules (for an excellent review on segmented plant viruses read [53]). Segmented viruses package their genome into one or more virions. For example, the genomes of cowpea chlorotic mottle virus (CCMV) and brome mosaic virus (BMV) (both from the *Bromoviridae* family) are divided into three gRNA molecules. Furthermore, one extra RNA molecule, called the subgenomic RNA (sgRNA), is also packaged. RNAs 1 and 2 are packaged in different capsids, while RNAs 3 and 4 (the sgRNA) are co-packed in a 1-to-1 ratio in the same capsid.

The structural simplicity of some plant viruses and bacteriophages has allowed us to in vitro assemble them in a test tube, from purified components, into infectious particles [64,65,66,67]. While sometimes criticized as reductionist and/or biologically irrelevant, in vitro approaches have been crucial to the understanding of the physical principles that control the assembly of empty virions and nucleocapsids, as well as RNA packaging [5,6,7,8,11,12,31,34,35,36,37,68,69,70,71,72,73,74,75,76,77,78,79,80,81,82,83,84]. These experiments have demonstrated that: (i) the assembly of (+)ssRNA viruses requires protein-protein and RNA-protein interactions, (ii) this process is spontaneous, and (iii) virion assembly and RNA packaging are coupled with each other. Furthermore, they have allowed us to use viruses as biotechnological tools in material sciences [85,86], chemistry [87,88,89,90] and nanomedicine [91,92,93,94,95,96].

In vitro systems have been extremely useful to understand assembly and packaging, both in the presence [34,35] and absence of specific RNA-CP interactions [5,68]. On the one hand, Comas-Garcia, M. et al. showed that for CCMV, in the absence of specific RNA-CP interactions the length of the packaged RNA strongly contributes to the packaging efficiency [68]. In other words, when a series of viral-derived truncated RNAs competed against the wild-type (WT) length RNA for a limiting amount of CCMV CP, the RNAs with a length similar to that of the WT had a higher packaging efficiency than RNAs that deviated from the WT length. This finding is in agreement with experiments in cultured cells [97,98] and in protoplasts [40]. On the other hand, Borodavka, A. et al. showed that for MS2 and satellite tobacco necrotic virus (STNV), selective packaging of the gRNA depends on specific gRNA-CP interactions [34]. These interactions are mediated by a series of RNA elements similar to the PS that are distributed across the gRNA (PS-like elements). PSs are usually defined as a unique structure/sequence in the gRNA, which is small compared to the length of this molecule, that has an extremely high-affinity for the CP [40,99,100,101]. The work from the Stockley, Twarock and Dykeman groups have demonstrated, by using in vitro and mathematical approaches, that in some cases selective packaging requires not only the “canonical PS”, but a series of PS-like elements, completely changing the PS paradigm [9,34,36,102,103,104,105,106,107,108].

The original concept of a high-affinity binding mechanism between the PS and the CP is not always consistent with some in vitro and in vivo data [109]. For example, it has been proposed that PSs may act as nucleation sites [109,110,111,112,113,114,115] or conformational switches [55,82,116] (see Figure 1). More importantly, there are other factors, aside from a PS (and/or multiple PS-like elements), that need to be taken into account. In particular, theoretical calculations and molecular dynamic simulations have shown that the overall size of the RNA (3D size) contributes to the ability of an RNA to be packaged [18,19,20,21,117,118,119,120,121,122,123]. These models are consistent with some in vitro [68] and in vivo [40] findings.

How an RNA is packaged during virion assembly is not a simple story. Biochemical, virological and structural data suggest that RNA packaging is far more complex than a scenario that depends exclusively on the presence of a PS. For example, the RdRp is not just the enzyme that synthesizes the viral RNA. This enzyme, or some of its subunits, can remodel different organelles and membranes (for more details see [46] and [47]). Membrane remodeling segregates the gRNA from the lumen of the rough endoplasmic reticulum by forming vesicles. This segregation seems to be crucial for flaviviruses (e.g., Hepatitis C and Dengue viruses) [124]. In BMV this segregation was initially observed in a yeast-based system [125]. However, Chaturvedi, S. et al. showed that the role of BMV RdRp on RNA packaging selectivity is not due to gRNA compartmentalization, but to a direct interaction between the RdRp and the CP [38].

## 3. Packaging Signals

Packaging signals (PSs) have long been associated with the specific incorporation of the genome of (+)ssRNA viruses, retroviruses and hepadnaviruses. While retroviruses and hepadnaviruses switch their genome from ssRNA to dsDNA, the molecule that is packaged during assembly is ssRNA; thus most of the packaging-related aspects described for (+)ssRNA viruses are applicable to these two types of viruses.

Packaging signals have been studied in ssRNA bacteriophages (e.g., *Leviviridae* family) [34,126], animal viruses (e.g., *Caliciviridae* [127,128], *Togaviridae* [98,129,130,131,132,133] and *Coronaviridae* [134] families) and plant viruses (e.g., *Bromoviridae* [135,136,137] and *Tombusviridae* [37,40] families). Most of the virological data from cultured cells are for animal viruses. PSs in ssRNA bacteriophages have been mostly studied in vitro. Packaging in plant viruses has been studied both in vivo and in vitro, although rarely for the same virus.

One of the best-known PSs and assembly processes is that of tobacco mosaic virus (TMV) [67,112,113,114,115,138,139]. In the absence of a nucleic acid and under non-physiological conditions, TMV CP can be in vitro assembled into structures similar to the native capsid [140,141,142]. This indicates that neither the presence of the PS nor the RNA is an absolute requirement for in vitro capsid assembly. However, the fact that capsids can assemble without RNA under solution conditions that are very different from the physiological, suggests that CP-CP interactions must be altered to overcome the absence of a nucleating template (i.e., the PS or the RNA). TMV can be assembled in the presence of heterologous RNA, however this process is far less efficient that in the presence of the homologous TMV RNA [139]. Similar results were observed when comparing the assembly of TMV CP in the presence of the polyA, polyI and homologous RNA [143,144]. These results are in agreement with a mechanism in which the PS is a nucleation site for capsid assembly, and this element lowers the activation energy of this process. In fact, disruption of the PS is sufficient to inhibit in vitro assembly [145]. For example, the binding of a family of small organic molecules (antofine analogs) to the PS is sufficient to inhibit this process [146]. This drug-mediated inhibition phenomenon supports a scenario in which in vitro TMV assembly is highly dependent on the presence of a PS [147].

Sorger and co-workers used in vitro disassembly experiments to identify the PS of turnip crinkle virus (TCV) [37]. They found that a small number of CPs, compared to the 180 CPs that make up the virion, had a much stronger interaction with a small RNA sequence (located in the RdRp gene) than with the rest of the TCV genome. Sorger and co-workers proposed that this small RNA sequence was the packaging signal [37]. In a series of very elegant experiments Qu and Morris narrowed down the TCV PS to 186 nt by measuring the ability of different deletion mutants to be packaged in vivo (i.e., protoplast) [40]. However, this PS is located in the CP and not in the RdRp gene (as proposed by Sorger et al. [37]). They inserted the TCV PS into the tomato bushy stunt virus (TBSV) gRNA (chimeric RNA) so that they could carry out head-to-head competition experiments, in the same protoplast, between the chimeric TBSV (+TCV PS) and TCV gRNA. These experiments showed that the insertion of TCV PS into the TBSV gRNA promoted packaging of the non-cognate RNA by the TCV CP (see Figure 2a). They also showed that the packaging efficiency of the chimeric RNA also depends on the length of the RNA; as the length of the chimeric RNA approached that of the WT, the packaging efficiency increased. This in vivo length dependence is in agreement with in vitro experiments on CCMV [68] and with Aura virus in cultured cells [98].

The Schlesinger group showed that a 132 nt-long fragment from the genome of the alphavirus Sindbis (SINV) had a high-binding affinity for the SINV CP [101]. A few years later, Frolova, E. et al. showed that this sequence increased SINV RNA packaging efficiency in infected cells [130]. As in most viruses, studying the PS in alphaviruses is complicated; insertions or deletions in this region disrupt the coding of the non-structural protein 1 (nsP1). To solve this problem, they inserted different sequences of the gRNA into the sgRNA (with the exception of Aura virus, alphaviruses (*Togaviridae* family) do not package their sgRNA [98]). Their goal was to find a sequence that promoted the packaging of the sgRNA. Insertion of the PS, identified in vitro by the Schlesinger group, into the sgRNA was sufficient to package it, thus validating the biological significance of this PS (see Figure 2b). In that same work they showed that Ross River virus (RRV), also an alphavirus, has three different fragments that enhance packaging of the sgRNA (three-part motif). These fragments are localized within the nsP2 gene. The PS located between nt 2902 and nt 3062 made a stronger contribution to packaging than the other two sequences. Nonetheless, the packaging efficiency of the sgRNA increased by inserting any of these fragments. The PS of Semliki Forest virus (SFV) is located almost in the same place as the second RRV PS. Interestingly, the PS for SFV, which is phylogenetically closer to RRV than to SINV, was identified as a single sequence (as in SINV) and not as a three-part motif (as in RRV) [133,148,149,150]. The differences between SFV and RRV PSs are a clear example of why we should not extrapolate what we learn from one virus to all members of its family (or even to the same clade).

Alphaviruses can be divided into SINV and SFV clades. By using a bioinformatic approach the Frolov group proposed that the PS for the SINV clade (e.g., Venezuelan encephalitis virus (VEEV)) is located in the nsP1 gene [129]. Also, they proposed that for the SFV clade (e.g., chikungunya virus (CHIKV) and RRV) the PS is in the nsP2 gene. Interestingly, they did not find that the secondary and tertiary SFV PSs, previously identified by the same group, were present in other members of this clade. To further characterize the PSs of different alphaviruses they constructed a series of chimeric genomes in which the structural proteins of either VEEV or SINV were replaced by those of SINV, CHIKV or VEEV, thus creating VEEV/SINV, VEEV/CHIKV and SINV/VEEV chimeric viruses. For example, the VEEV/SINV chimera has the non-structural proteins of VEEV and the structural proteins of SINV. All of these chimeras had an infectivity similar to that of each WT virus. This result could be interpreted as all of these viruses sharing the same PS. However, this interpretation is not consistent with their bioinformatic data; the putative packaging signal for the VEEV and SINV/CHIKV are located in different genes (i.e., nsP1 and nsP2, respectively). Furthermore, this “high-efficiency cross-packaging” could suggest that viruses that diverged thousands of years ago have the same PS. Such a suggestion seems unlikely, especially after taking into consideration that the N-terminal domain of the CP (responsible for binding to the gRNA) differs greatly between alphaviruses [151]. Unfortunately, their assays were based on measuring the infectivity of each chimeric virus on its own. In the absence of the WT RNA, the chimeric RNA competes only against cellular RNAs and the sgRNA, but not against the molecule that has evolved to be packaged by the CP (i.e., WT gRNA). In fact, similar studies with chimeric SINV/RRV viruses confirm the cross-packaging observed by Kuhn, R.J. et al. [152]. Although in a very different context, it was shown that it is essential to establish a proper competition in order to measure the in vitro packaging efficiency of CCMV CP [68]. Similarly, without the proper competitor RNA, it is almost impossible to measure binding specificity between the HIV-1 structural polyprotein, Gag, and its PS [153,154]. A possible explanation for the “cross-packaging” observed with chimeric alphaviruses will be discussed in the last section.

As mentioned before, studying the role of a PS during a viral infection is complicated. The Stockley and Twarock groups came up with an elegant methodology to find PSs by using an in vitro approach [34,36,84]. First, by doing SELEX experiments (Systematic Evolution of Ligands by Exponential Enrichment; for more information about SELEX read [155]) with a large library of RNA aptamers they were able to select a small set of sequences with a high-binding affinity and specificity for a given CP. Second, Twarock and Dykeman developed a mathematical approach to find RNA elements within the genome of interest that have a similar motif as the RNA aptamer chosen during the SELEX step. Finally, they did single-molecule fluorescence correlation spectroscopy (sm-FCS) to in vitro assemble different CPs (i.e., STNV, MS2 and HBV) to test these putative PSs. By using sm-FCS they were able to work under extremely diluted CP and RNA concentrations, thereby minimizing non-specific protein-RNA interactions that could drive non-selective packaging of STNV, HBV and MS2 viral RNAs (they proposed that these concentrations are similar to those found in infected cells and thus biologically relevant). They showed that only RNAs containing the PS assembled into stable virus-like-particles with the right structure, while PS-minus and non-cognate RNAs formed misassembled products that are unstable and disassemble in the presence of RNase A. These experiments allow them to propose that selective packaging does not depend on one PS, but on a series of PS-like elements distributed along the gRNA. They suggested that a distribution of PSs and PS-like elements can better modulate virion assembly in comparison to a process that depends on one specific RNA segment. Furthermore, it might reduce the probability that a few mutations would disrupt packaging. It is important to point out that the biological activity of these PS and PS-like structures during RNA packaging in an infected cell is yet to be determined. Nonetheless, there is some available data to start putting things into perspective. For example, experiments with HCV in cultured cells showed that the PS identified in vitro by SELEX experiments contribute to RNA packaging [108]. On the one hand, there is independent evidence that the PS for HIV-1 is not a single stem-loop (as originally proposed [100,156]) but a series of unpaired Guanosyl residues distributed along the 5´UTR and Gag open reading frame [110,157,158]. This type of PS is consistent with the multiple PS-like scenario. On the other hand, the Lomonossof group has strong evidence that mutations that should disrupt the in vitro-identified STNV PS have no effect on in vivo packaging, thus questioning the biological role of these PSs for this particular virus [159].

One possible explanation for the apparent lack of specificity (i.e., “cross-packaging”) observed in the experiments from Kim, D.Y. et al. [129] and Kuhn, R.J. et al. [152], as well as the discrepancy between the in vivo and in vitro experiments in STNV, could be that in the absence of a PS or PSs, viral RNAs have a secondary mechanism that gives them an advantage over cellular RNAs. It is plausible that RNA replication and/or co-translational packaging play a role in the selectivity of this process. The evidence to support this hypothesis is explained in the following section.

## 4. Replication and Translation Contribute to Packaging Specificity

As we have extensively discussed, the selective packaging of the gRNA is often attributed to the presence of a PS and/or multiple PS-like elements. However, there is sufficient evidence that RNA replication, by the RdRp, is necessary for gRNA packaging of poliovirus (*Picornaviridae* family) [43], Kunjin (a subtype of west Nile virus from the *Flaviviridae* family) [42], Flock house (*Nodaviridae* family) [41,97,133,160], Venezuelan equine encephalitis (*Togaviridae* family) [39,161] and brome mosaic (*Bromoviridae* family) [38,44,162] viruses. In this section we will briefly describe some findings that support the importance of RNA replication for selective packaging in these viruses.

Nugent and co-workers used a replication-competent poliovirus RNA to investigate whether RNA replication by an active RdRp is required for selective packaging [43]. This RNA had deletions in the capsid gene (replicon RNA); thus, it could only be packaged in the presence of an RNA that contains the structural proteins (helper RNA). A replicon is an RNA that has all the genes required for replication. A helper RNA provides the structural proteins required to package the replicon RNA. Furthermore, in these experiments the replicon cannot be replicated in the presence of guanidine, while replication of the helper RNA is insensitive to this small molecule. By determining the RNA content of virions assembled in the absence and presence of guanidine, that is in the presence and absence of replication, they were able to determine that only RNAs that can be replicated by the RdRp are packaged. This elegant system allowed them to conclude that RNA replication is most likely responsible for poliovirus RNA selective packaging in infected cells.

By using a slightly different approach Khromykh, A. A. et al. showed that the packaging of Kunjin RNA is coupled to RNA replication [42]. To demonstrate this, they transfected cells with two different DNA-based viral constructs: a WT and a nonreplicating clone (deletion in the NS5 gene of the RdRp). It is important to remember that the transfection of cells with cDNA clones of (+)ssRNA viruses produces two sources of viral RNAs: nuclear (transcripts from the cDNA) and cytoplasmic (amplification of the gRNA by the RdRp). Transfection of cells with the WT clone resulted in the assembly of particles that contained only viral RNA. Particles produced with the nonreplicating clone contained only cellular RNAs. It should be pointed out that in the absence of the RdRp the amount of nonreplicating RNA is considerably lower than for the WT RNA or when the transfected cells constitutively express the RdRp (i.e., stable cell lines that express the RdRp). Thus, in this case the viral RNA is not one of the predominant species. Transfection of cells that constitutively express the RdRp with the nonreplicating virus produced particles that contained viral RNA. Khromykh, A. A. and co-workers concluded that RNA replication is required for packaging of the viral RNA.

Coupling between RNA replication and packaging has been observed in Venezuelan equine encephalitis virus (VEEV) [39]. By using a VEEV-based replicon/helper system Volkova, E. et al. demonstrated that selective packaging of a defective helper RNA (DH RNA) not only depends on the presence of a PS, but on its ability to express nsP1-3 (three of the four proteins that make up the RdRp). An alphavirus replicon RNA is a viral-derived RNA molecule that contains all the genes required for RNA replication (open-reading-frame 1 or ORF1). To test if RNA replication and packaging are coupled with each other Volkova, E. et al., deleted the structural genes (ORF2) of the replicon RNA. This molecule cannot produce viral particles and therefore is noninfectious; thus, to generate viral particles a DH RNA was supplied in trans (i.e., in a different RNA molecule). A DH RNA is a viral-derived molecule that encodes an inactive RdRp (ORF1) but contains a functional ORF2. In general, the ability of a DH RNA to be packaged depends on the presence of the ORF1, which contains the genes required for RNA replication and also the putative PS. Only when the replicon and the DH RNAs are present in the same cell can the produced particles contain viral RNA. By using this system, they concluded that the presence of nsP1-3 is essential to make the DH RNAs persist among the population of viral RNAs produced during the experiments (i.e., replicon RNA, (−)ssRNA, sgRNA and DH RNAs) and thus to be packaged. In other words, selective packaging of VEEV RNA requires a PS and RNA replication. Nonetheless, it should be mentioned that there is some evidence that packaging of defective interfering RNAs in SFV (another alphavirus) seems not to depend on RNA replication [133]. These contradictory results between these two alphaviruses reinforce our concern regarding generalizing findings for one virus to all viruses.

It is possible that when RNA replication (at least for poliovirus, VEEV and Kunjin virus) is inhibited, cellular RNAs are packaged because they are far more abundant than the viral RNA. This scenario is not unique to (+)ssRNA viruses. It has been observed that when HIV-1 and Moloney murine leukemia virus Gag proteins (the structural polyprotein in retroviruses) are expressed in cells in the absence of the gRNA [163], cellular RNAs are packaged. Furthermore, the packaging efficiency of the cellular RNAs is proportional to their concentration in the cell. In other words, for some viruses it is possible that in the absence of a PS and replication (by the RdRp) packaging efficiency depends on RNA concentration.

Another possibility that involves RNA replication during RNA packaging is a “three-body interaction” between the RdRp, gRNA and the CP. The Rao group has had a long-time interest in studying the relationship between RNA replication and packaging in Bromoviruses and Nodaviruses [38,45,164,165,166,167,168]. The two most studied viruses from the *Bromoviridae* family belong to the *Bromovirus* genus: brome mosaic virus (BMV) and cowpea chlorotic mottle virus (CCMV). A distinctive characteristic of these viruses is that their genome is divided into three molecules. RNA 1 and 2 code for the two proteins that constitute the RdRp. RNA 3 has two ORFs: the first one codes for the movement protein and the second one for the CP. Only the movement protein is translated from RNA 3. During minus to plus-strand synthesis there is a moment in which the RdRp binds to the intergenic promoter and synthesizes an RNA that contains only the ORF2 of RNA 3, thus making the sgRNA. The CP is not translated from RNA 3 but from the sgRNA (RNA 4). In order to understand some of the general features of these two viruses Allison, R. F. et al. infected plants with CCMV and BMV in vitro transcripts [162]. They showed that heterologous infection of a plant with RNAs 1 and 2 from one virus (e.g., CCMV) and RNA 3 from the other virus (e.g., BMV) is sufficient for RNA replication. Furthermore, they observed heterologous packaging; for example, RNA 1 and 2 from CCMV were packaged by the BMV CP and the other way around. In fact, cross-packaging between CCMV CP and BMV RNAs has been observed in vitro [5,68]. On the one hand, it is possible that heterologous packaging depends on the ability of an RNA to be replicated by the RdRp. This implies that CCMV and BMV may not require a PS and that packaging depends entirely on the ability of an RNA to be replicated. This is not surprising as there is an excess of RdRp molecules in an infected cell. On the other hand, it is possible that these two viruses share the same PS. However, it has been shown that the efficient assembly of BMV, but not of CCMV, requires a PS [44,53,68,135,136,137,169]. In vitro assembly of BMV requires a PS that is located at the 3´UTR of all four BMV RNAs [135]. This is a 200-nt long highly conserved sequence that can be folded into a tRNA-like structure (TLS). Deletion of this region suppresses in vitro assembly with any of the four BMV RNAs. Interestingly, addition of the TLS in trans (i.e., as a separate molecule) rescues in vitro assembly [170]. Furthermore, addition of non-viral tRNAs is sufficient to promote efficient virion assembly [170]. This suggests that the 3’UTR TLS is indeed a PS. However, unlike BMV, CCMV does not require a TLS for efficient in vitro assembly [68]. The different in vitro assembly requirements between CCMV and BMV suggest that the in vivo cross-packaging observed between these two viruses is not due to a common PS.

A few years later, Annamalai, P. et al. used an *Agrobacterium tumefaciens* system to infect plants with BMV cDNAs and decouple replication from packaging [171]. They showed that in the absence of RNA replication (i.e., expression of the sgRNA that encodes the CP, but not RNAs 1, 2 and 3) packaging of the sgRNA was not specific: cellular RNAs and the sgRNA were packaged to levels comparable with each other. The presence of active replication or a functional RdRp rescued the packaging specificity of the sgRNA. Perhaps the most interesting result was that efficient packaging of the sgRNA not only depended on RNA replication but also on translation.

How does RNA replication contribute to selective packaging? As we have already mentioned, the simplest mechanism is that in which the RdRp increases the local concentration of the viral RNAs so that it dominates the RNA population. Nonetheless, this is not the only possibility. Chaturvedi, S. and Rao, A.L.N. carried out a series of very impressive experiments to test the hypothesis that genome packaging specificity requires a physical and direct interaction between the RdRp and CP [38]. By using a Bimolecular Fluorescent Complementation (BiFC) approach they characterized the interactions in planta of BMV RdRp proteins 1a (p1a) and 2a (p2a) (together they make the RdRp). They concluded that BMV CP has a physical and direct interaction with p2a but not with p1a and proposed that this interaction is necessary for packaging specificity.

Flock house virus (FHV) is a bipartite (two-molecule segmented) insect virus from the *Nodaviridae* family. Unlike in Bromoviruses, both gRNAs (RNA 1 and 2) are packaged into a single virion [97]. By using a baculovirus system Krishna, N. K. et al. showed that specific packaging of RNA 2 requires RNA replication by the RdRp [97]. As in the case of the Kunjin virus cDNA system, this baculovirus approach produces two different populations of RNAs: one from transcription of the cDNA and one from RNA replication by the RdRp. They proposed that perhaps the RNA synthesized by the RdRp is likely to be present in a more suitable conformation for packaging than RNA synthesized by the cellular RNA polymerase. They also made the point that when the RdRp is present in the cell the levels of viral RNA are greater than when the RNAs are synthesized by the RNA polymerase in the nucleus. A few years later the same group showed that neither RNA 1 nor 2 were efficiently packaged if the CP was supplied in trans from a nonreplicating RNA (i.e., as an independent mRNA) [41]. Later, they expanded these types of experiment and demonstrated that particles assembled from CP translated from a replicating RNA contained only viral RNA, while CP translated from nonreplicating RNAs produced particles that contained cellular RNAs [160].

So far it is clear that PSs are necessary but not sufficient for most viruses. In such cases, replication and/or translation are required to make a viral RNA suitable for packaging. As it turns out, RNA replication modifies cellular membranes. This rearrangement is mostly caused by the viral non-structural proteins; therefore, we will now discuss the role of such cellular modifications on packaging specificity.

## 5. Membrane Re-Arrangement and Viral Factories

RNA replication of (+)ssRNA viruses that infect eukaryotic cells results in virus-induced membrane remodeling [46,47]. The remodeled structures have been designated as virosomes, virus inclusions, virus factories, viral replication factories, cytopathic vacuoles or viroplasm. Sometimes they are incorrectly called inclusion bodies, a term borrowed from aggregates observed in bacteria.

Virus factories have been described as spherule invaginations (FHV [51], SFV [172] and BMV [172]), rossettes (poliovirus [173]), double-membrane vesicles (HCV [124,174,175]) and convoluted membranes (DENV [124]). Paul and Bartenschlager proposed that viral replication factories induce one of two different membrane alterations: invaginations/spherules and double membrane vesicles [176]. Generally, the endoplasmic reticulum is the most preferred membrane; however, the mitochondria [51], lysosomes [172,177], peroxisomes [178] and chloroplasts [179] can be used as viral factories. It is not clear if these replication sites are always associated with nucleocapsid assembly, although viruses have been observed inside these structures [172,179,180]. These factories are thought to have different functions [176], namely: (i) increase the local concentration of the components required for replication and assembly, (ii) anchor structural proteins and/or the RdRp complex, (iii) protect dsRNA intermediates from the host defense mechanisms and (iv) provide sites for virus assembly. Furthermore, they are responsible for the spatial and temporal coordination of translation, RNA replication and assembly (Figure 3).

The assembly factories of the Togaviruses SFV, SINV and Rubella virus (RUBV) are localized around endosomes and lysosomes and create cytopathic vacuoles (CPV). In these cases, the remodeled membranes act as assembly sites. From the work by Zhao, H. et al. it is obvious that assembled SFV virions can be found inside endosomes [181]. As in SFV, infection with RUBV modifies the endosomes and lysosomes to create CPVs [177]. In the early time points after a SINV infection spherules are observed at the plasma membrane. These are later internalized and assembled into CPVs [182]. So far, there are three important observations: membrane remodeling requires the nonstructural proteins, the RdRp is anchored to the cytoplasmic surface of the CPV, and the ER is in close proximity to the CPV [176,180,182,183]. While there is yet no direct evidence that Togavirus packaging specificity requires the assembly of CPV, the presence of the RdRp and virions in the CPVs, along with the requirements of an RNA to be replicated and translated for efficient packaging suggest that the CPVs contribute to specific RNA packaging.

BMV ER-derived invaginations are the result of the interaction between the nonstructural protein p1a and cellular proteins [50]. This association leads to a recruitment of the viral RNA into the spherules via an interaction between the viral proteins p1a and p2a [184]. Once this complex is assembled RNA replication is initiated. Interestingly, it was first reported that BMV CP was not localized inside these invaginations [52,184]. However, as Bamunsinghe et al. pointed out, this observation is hard to reconcile with a replication-coupled packaging mechanism in which virus assembly depends on viral factories [45]. To understand this disconnection Bamunsinghe et al. used immunofluorescence confocal microscopy and transmission electron microscopy to explore the subcellular localization of BMV CP synthesis [45]. They found that CP localizes in the ER and overlaps with the viral replication sites. Unlike Togaviruses, expression of CP alone in *N. bethamiana* is sufficient to produce ER-derived invaginations. However, the role of the viral replication factories in BMV selective packaging remains unclear. Based on the fact that the p1a and p2a are present in the invaginations and that a direct interaction between p2a/CP is required for specific packaging, one could conclude that membrane remodeling is necessary for the selective packaging of BMV gRNA. However, Schwartz et al. found that, at least in a yeast system, only RNA 3 is inside the spherules [184]. The lack of association of RNAs 1, 2 and 4 with the spherules suggests that selective packaging of BMV RNAs is not directly linked with membrane remodeling. Nonetheless, it has been shown that vesicle induction in planta promotes cell-to-cell viral movement [185]. It is possible that membrane remodeling is not only related to the assembly RNA replication sites but might be required for other purposes. However, these contradictory results are far from being resolved and require further investigation.

## 6. Conclusions: Finding a Common Ground

The gRNA of (+)ssRNA viruses is like a Swiss Army knife in that (i) it contains the minimal genetic information required for the infectious cycle, (ii) is the template for RNA replication, (iii) is a messenger RNA, (iv) can act as a scaffold during virion assembly, and (v) has to be able to present different RNA structures to regulate these processes during the infectious cycle. On the one hand, it has to have an architecture similar to cellular RNAs so that it can be translated, evade the protective cellular response against dsRNA and interact with host-cell factors. On the other hand, it has to be different enough from the cellular RNAs to be identified by the RdRp and selectively packaged during assembly.

It is important to keep in mind that assembly and RNA packaging occur in the presence of a large and diverse population of cellular and viral-derived RNAs, yet only the gRNA is packaged (with a few exceptions like Aura virus and retroviruses). This observation leads to a series of questions: *what makes the gRNA different from cellular and viral-derived RNAs so that only the full-length viral RNA is packaged? How does a packaging signal work? Are packaging signals sufficient for packaging selectivity*? *What other factors, aside from a packaging signal, contribute to selective packaging?*

The first question is most commonly answered by attributing selective packaging to the presence of a single RNA element that is unique to the gRNA and that establishes highly specific interactions with the viral structural proteins. This has been the working hypothesis for most (+)ssRNA viruses and retroviruses. However, based on the evidence presented here this scenario is not always precise. An improvement to this view is that for some viruses there is not one PS, but of a series of PS-like elements [9,105,107,186]. One biological advantage for this alternative hypothesis is its resilience to mutation. If packaging of the gRNA were to exclusively depend on a few bases that make up a highly regulated structure, then mutations on or around the PS could easily render a virus inactive. However, by having a series of PS-like elements distributed across the gRNA, mutations on the main PS (or on any of the PS-like elements) would have a weaker negative effect than when there is only one PS.

The data presented here suggest that there is no universal PS-mediated packaging mechanism (see Figure 1). It is generally accepted that a PS is an RNA element with high-binding affinity for the viral structural proteins, but this is not always the case. For example, it was thought that the HIV-1 PS (known as Ψ), located within the 5’UTR of the gRNA, controls selective packaging by establishing high-affinity interactions with the structural polyprotein Gag [187,188]. However, this mechanism is not consistent with the fact that under physiologically relevant salt concentrations the difference in binding affinities between Ψ and any other nucleic acid to Gag is very small (usually it is about a 2 to 3-fold difference) [109,187,188,189]. It was recently shown by in vitro binding assays that non-specific electrostatic interactions can be strong enough to mask specific Ψ-Gag interactions [154,189]. It is only when these non-specific interactions are blocked by increasing salt concentration, adding an excess of a competitor RNA or decreasing the basic character of Gag that Ψ presents binding specificity for this protein [154,189]. More importantly, by decreasing the relative strength of these non-specific interactions it became clear that assembly on HIV-1 Ψ is much more efficient than on a non-Ψ RNA. Comas-Garcia et al. proposed that HIV-1 Ψ acts by lowering the activation energy of assembly rather than by acting only as a high-affinity binding site [110]. In fact, experiments in cultured cells support the hypothesis that Ψ is the nucleation site for immature virion assembly [111]. There is evidence that the PS for the bacteriophage MS2 is not only a high-affinity binding site for its cognate CP, but it is also an allosteric switch that changes the assembly pathway to one with a lower free energy [190,191,192]. This switch provides the gRNA with an advantage over the cellular RNAs during virion assembly by lowering the Gibbs free energy of the assembly pathway. However, with a few exceptions the lack of knowledge of other PS-mediated packaging mechanisms is considerable. Solving this problem should be a priority in the field of (+)ssRNA virus assembly.

The renewed interest in PS has overshadowed the contribution of RNA replication, translation and compartmentalization to RNA packaging. It is clear that at least in Picornaviruses, Togaviruses, Nodaviruses, Flaviviruses and Bromoviruses, a PS is not sufficient to explain selective packaging. With the exception of Alphaviruses (*Togaviridae* family) there is no concrete evidence that these viruses utilize a PS in vivo (the activity of Bromovirus PSs has only been studied in vitro). The evidence supports the idea that the ability of the gRNA to be replicated by the RdRp is a key factor that determines if this molecule can be packaged. For example, the high degree of cross-packaging efficiency observed in alphaviruses by Kim et al. [129] and Kuhn et al. [152] strongly supports the hypothesis that in the absence of a “true PS” the ability of the RdRp to recognize and replicate the viral RNA controls packaging. Unfortunately, the role of the RdRp during packaging is not simple, it is involved in several processes and steps of the viral infectious cycle. In vitro biophysical and biochemical approaches could be used to decouple some of these processes and shed some light into the contribution of the RdRp to packaging specificity.

Other characteristics that may determine how an RNA is packaged is whether RNA replication produces other RNA species aside from the gRNA (e.g., sgRNA) and if more than one RNA molecule is packaged. For example, virus replication in Togaviruses (SINV and RRV), Betacarmoviruses (TCV), Nodaviruses (FHV) and Bromoviruses (BMV and CCMV) requires the synthesis of sgRNAs. Bromoviruses always co-package their sgRNA, while Betacarmoviruses, Nodaviruses and Togaviruses (with the exception of Aura virus) do not package this molecule. This difference suggests that these are secondary mechanisms to control packaging of the sgRNA. It is possible that the amount of packaged nucleotides is a key factor (see next paragraph). In the case of the Togaviruses the sgRNA does not contain the PS (again Aura virus being the only exception); therefore, it is likely that selective packaging of the gRNA could depend on the presence of a PS, as well as on the ability of an RNA to be replicated. It is not completely clear how sgRNA packaging is regulated in Bromoviruses, or how the sgRNAs in the Betacarmoviruses and Nodaviruses are excluded during packaging. On the one hand, in Bromoviruses RNA 3 and 4 (sgRNA) are about 2/3 and 1/3 the length of RNA 1, respectively. Therefore, by co-packaging RNA 3 and RNA 4 the total number of nucleotides in the virion is almost the same as capsids containing either RNA 1 or 2. On the other hand, the sgRNA of Nodaviruses has a mass that is about 9% of the RNA mass contained in the virion (RNA 1 + 2). Thereby, the sgRNA might be too small to be packaged. In other words, it is likely the sgRNA of Nodaviruses cannot be packaged because is too small compared to the other two RNAs and cannot assemble viruses with an RNA mass similar to the normal viral particles. This scenario is consistent with the in vitro packaging efficiency experiments with CCMV CP and truncated RNAs that showed that there is an optimal RNA length and number of total nucleotides required for high packaging efficiency [68].

Belyi and Muthukumar found that the genome size per capsid is directly proportional to the inner charge of the virion [193]. In other words, the number of packaged nucleotides depends on the number of basic residues in the interior surface of the nucleocapsid. This implies that the length and number of packaged RNA molecules inside an icosahedral (+)ssRNA virus is tightly regulated. In vitro and cultured-cell experiments with Bromoviruses, Alphaviruses, Nodaviruses and Betacarmoviruses have shown that the length of the packaged RNA contributes to the packaging efficiency. The optimal length/number of packaged molecules by these viruses is consistent with Bely and Muthukumar hypothesis. It is likely that RNA co-packaging (at least in Bromoviruses and Nodaviruses) is a way to satisfy the optimal number of packaged nucleotides. Nonetheless, we cannot rule out any sequence-specific interactions that could contribute to RNA packaging. The mechanisms by which the sgRNAs are either packaged or not, as well as RNA co-packaging, are still not well understood. In fact, the contribution of these processes to packaging selectivity is yet to be determined.

So far, no PS for Flaviviruses have been identified in cultured-cells experiments. However, the presence of RNA replication and assembly sites seem to be crucial for these viruses. It is plausible that selective packaging in Flaviviruses is primarily due to RNA replication/co-translational packaging within the assembly sites and not by a PS-mediated mechanism. Although, at this point this is only one of the possible explanations. Nonetheless, the CP has to be able to at least distinguish between the (+) and (−)ssRNA and thus a weak PS may be required for this selection.

Are there any viruses with a packaging mechanism that depend only on the presence of a PS? This is a very complicated question that perhaps cannot be fully answered. However, there are a few cases in which a PS can be the primary, if not the only, determinant for assembly. For example, TMV in vitro assembly occurs only the presence of a PS [67,112,113]. Furthermore, (+)ssRNA bacteriophages (e.g., from the *Leviviridae* family) are also likely candidates to fall into this category. The binding affinity of CP to PS for these viruses is in the low nanomolar regime (≈2.5–0.4 nM) [99,126]; the strength of this interaction is between 10 to 100-fold higher than for CCMV CP [16] or HIV-1 Gag [154]. One explanation for such strong PS-CP interaction could be the nature of the host. Viruses from the Leviviridae family infect bacteria, therefore RNA replication and assembly do not involve viral factories. This means that the CP has to be able to identify the viral RNA in an environment with a high concentration of cellular RNAs, without the help of compartmentalized assembly sites. It is plausible that the nature of the prokaryotic cell has exerted a strong evolutionary pressure in these viruses that resulted in a packaging mechanism that depends exclusively on the presence of a PS.

The molecular mechanism that controls selective packaging in (+)ssRNA viruses is very complex and depends on a series of interactions and characteristics that are specific to each species. There are instances in which some of these characteristics can be shared among viruses that belong to the same genus or family, or even between unrelated viruses. However, the specific details of these molecular mechanisms can differ greatly even between species that are evolutionarily very close to each other. Viruses can solve the same problem by using very different solutions.

Finally, the lack of knowledge of all the interactions that control packaging of the gRNA has led us to neglect this process as a possible target for antiviral therapies. Only a deep and comprehensive understanding of the similarities and differences between these processes among different viruses will allow us to develop novel therapies aimed at one of the most crucial steps of the viral infectious cycle. Remember, a virus that packages the wrong RNA is a dead end.

## Figures and Tables

**Figure 1 viruses-11-00253-f001:**
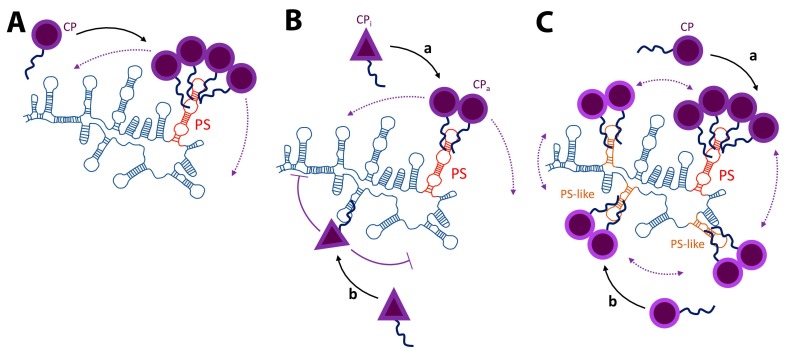
PS-mediated packaging mechanisms. (**A**) High-affinity binding mechanism. Selective packaging of the genomic RNA (gRNA) is mediated by high-affinity binding between the capsid protein (CP) and the packaging signal (PS). In this scenario the PS is one small RNA element (and/or sequence) present only in the gRNA. (**B**) PS allosteric mechanism. Binding of the CP to the PS is an allosteric switch that changes the conformation of the CP from one with weak CP-CP interactions (CPi) to one strong interactions (CPa). (**C**) Multiple PS-like elements. There is one PS and a series of PS-like elements distributed across the gRNA. The interaction between the PS and the CP (a) are stronger than between the PS-like elements and the CP (b). Unlike the first two scenarios, all these clusters contribute to the nucleation of virion assembly. In all cases the unstructured tail represents the CP N-terminal domain. For most (+)ssRNA viruses the N-terminal domain of the CP is extremely basic and is required for RNA/CP interactions and RNA packaging.

**Figure 2 viruses-11-00253-f002:**
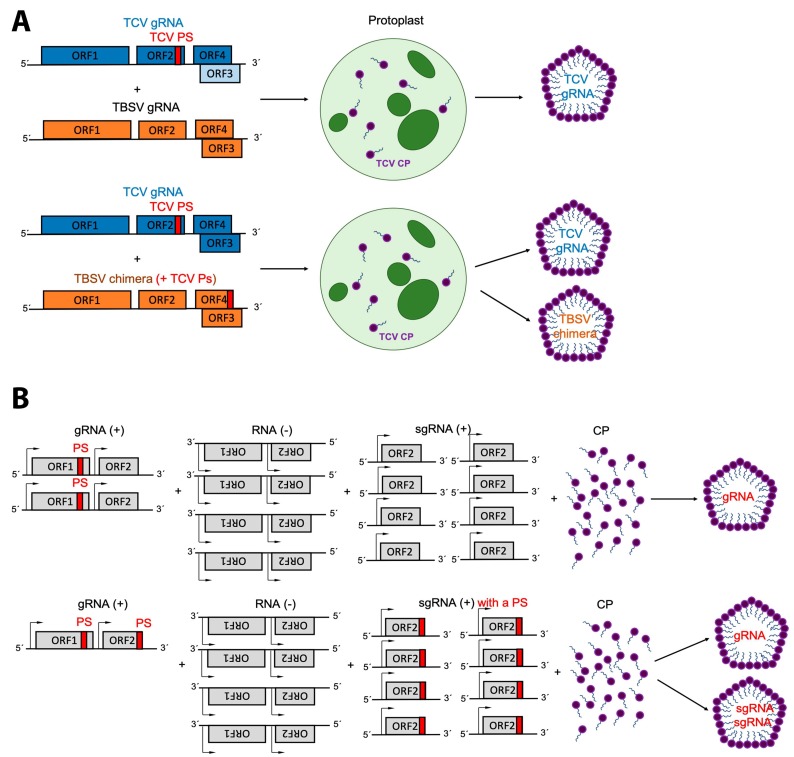
RNA replication for (+)ssRNA viruses and PS-mediated packaging. (**A**) Insertion of turnip crinkle virus (TCV) PS into tomato bushy stunt virus (TBSV) gRNA is sufficient for TCV capsid protein to package the TBSV RNA in protoplast and in the presence of the wild-type TCV gRNA. (**B**) In viruses such as Togaviruses, the sgRNA and the (−)ssRNA, which are in excess with respect to the gRNA, are not packaged. This is most commonly attributed to presence of a PS. For example, insertion of the Venezuelan equine encephalitis PS in its sgRNA is sufficient to package this RNA molecule. ORF refers to open reading frame.

**Figure 3 viruses-11-00253-f003:**
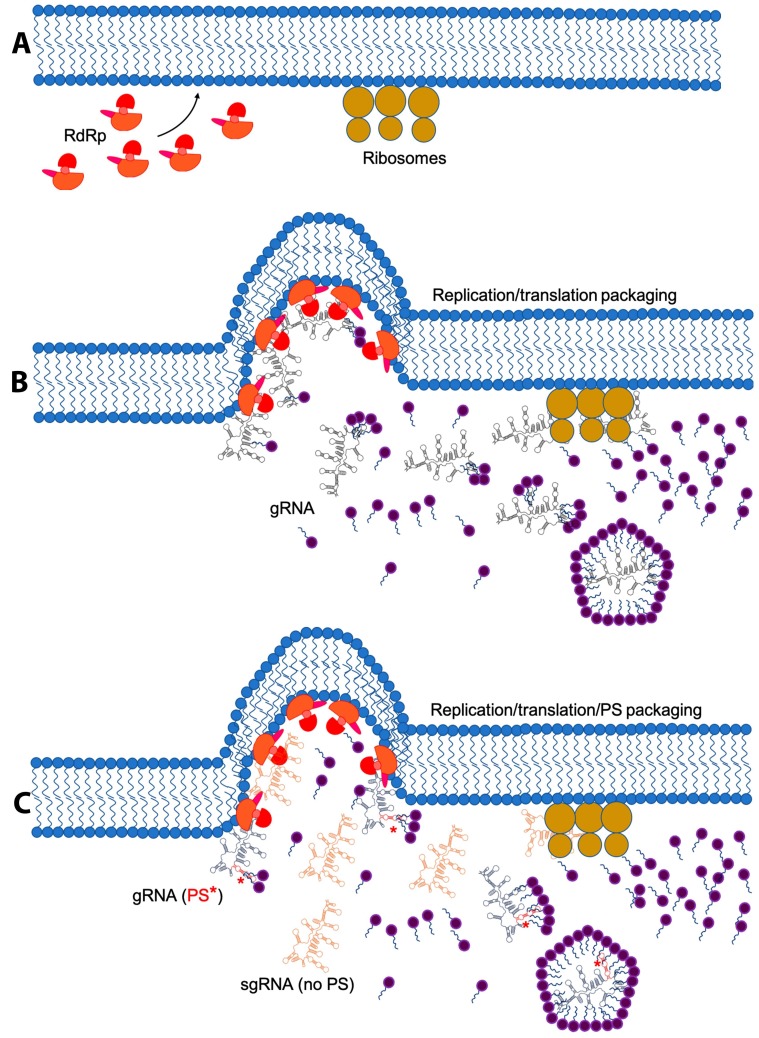
The role of RNA replication during RNA packaging. (**A**) Most RNA-dependent RNA polymerases (RdRp) of (+)ssRNA viruses interact with organelle membranes, creating viral assembly sites or replication factories. Sometimes these sites are very close to the ER. (**B**) There are viruses in which selective packaging seems to depend only on the ability of an RNA to be replicated by the RdRp (e.g., poliovirus). There is evidence that the ability of an RNA to act as a template for translation contributes to packaging selectivity (e.g., flock house virus). (**C**) In some other cases, the contribution of RNA replication is not sufficient, and the presence of a PS (red element with an *) is required for selective packaging. For example, in alphaviruses the presence of this signal is key to package only the genomic (blue with the PS in red with an *) but not the subgenomic RNA (orange).

**Table 1 viruses-11-00253-t001:** Summary of some of the interactions required for selective packaging of the gRNA. This process can require a PS, assembly/replication sites (viral factories) and/or RNA replication by the RdRp. In some cases, it is not known if there are PS. Also, the contribution of some of these factors has not been determined. * Refers to experiments done in cultured cells or protoplasts, ** indicates experiments done in vitro with reconstituted components, N.D., not determined and CPV means cytopathic vacuoles (see Section 4).

Family	Genus/Clade	Species	Packaging Signal-Mediated Packaging	Replication-Mediated Packaging*	Assembly/Replication Sites	Length-Dependence
*Togaviridae*	Alphavirus/SINV	Sindbis (SINV)	One (nsP1 *)	N.D.	CPV	Yes *
	Alphavirus/SINV	Venezuelan equine encephalitis virus (VEEV)	One (nsP1 *)	Yes	CPV	N.D.
	Alphavirus/SFV	Semliki forest virus (SFV)	One (nsP2 *)	No	CPV	N.D.
	Alphavirus/SFV	Ross river virus (RRV)	Three (nsP2 *)	N.D.	CPV	N.D.
*Bromoviridae*	Bromovirus	Brome mosaic virus (BMV)	One (3′UTR **)	Yes	ER	N.D.
	Bromovirus	Cowpea chlorotic virus (CCMV)	None **	Yes	N.D. (Probably ER)	Yes **
*Nodaviridae*	Alphanodavirus	Flock house virus (FHV)	N.D.	Yes	ER	Yes *
*Tombusviridae*	Betacarmovirus	Turnip crinkle virus (TCV)	One (CP */RdRp **)	N.D.	N.D.	Yes *
*Picornaviridae*	Enterovirus	Poliovurus	N.D.	Yes	Golgi	N.D.
*Flaviviridae*	Flavivirus	Kunjin virus (KUNV)	N.D.	Yes	ER	N.D.
*Flaviviridae*	Hepacivirus	Hepatitis C (HCV)	Multiple **	N.D.	ER	N.D.
*Hepadnaviridae*	Orthohepadnavirus	Hepatitis B (HBV)	Multiple **	N.D.	N.D.	No **
*Leviviridae*	Levivirus	Enterobacteria phage MS2	Multiple **	N.D.	No	No

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
