# Peer review of "Packaging of Genomic RNA in Positive-Sense Single-Stranded RNA Viruses: A Complex Story"

_viruses, 2019, doi:10.3390/v11030253_

Round 1

Reviewer 1 Report

The major focus of this review is to summarize the packaging signals in viruses with having single-strand, positive sense RNAs.  Although it is valuable information and deserves publication in Viruses, it would have been easy to follow the review had the authors divide the review into two three sections: Viruses with monopartite, bipartite and tripartite genomes (both bipartite. Most importantly, the authors should expand and discuss in more detail the first packaging signals identified in iconic virus TMV. Also, the authors ignored to discuss tRNA-mediated assembly in BMV. After addressing these issues, the authors should pay attention to the following suggestions as well to improve the overall quality of the review.

Page 2, line 69: “….to package only the right RNA”.  The use of the word “right” is colloquial and must be substituted with more appropriate scientific work such as “… “to package only the progeny RNA having optimal sequence/structural information”.

Page 4, lines127-130: “ the genome …are divided into three gRNA molecules and one subgenomic RNA (sgRNA)….”. This is wrong. In virology, by definition the term genome is referred to those RNAs that are required to initiate infection. Consequently, RNAs that are not part of the genome are referred to as sgRNA. Please re-write the sentence.

References #159 and #161 are the same. Delete one.

Author Response

1)      The reviewer suggested to divide the article by sections that separate viruses according to whether the genome of the virus is monopartite or segmented. We believe that this division, although reasonable, could defeat the purpose of the article. Our intention is to focus on the different processes that contribute to (+)ssRNA packaging. One of the key points is that some of these processes are shared between monopartite and segmented viruses. Thus, the intention of the current division is to point out that viruses that are very different from each other can share common mechanisms. Furthermore, the suggested division could lead to a fair amount of redundancy, making his article hard to follow.

2)      We have expanded the discussion about TMV packaging signal, see page 5 lines 209-223. We also realized that we forgot to cite some of the original experiments from Fraenkel-Conrat and Singer (page 5, lines 205-206).

3)      We have added the discussion about tRNA-mediated BMV assembly, see page 11 lines 424-432.

4)      Finally, we incorporated all other suggestions.

Reviewer 2 Report

Dr Comas-Garcia provides a review on the RNA packaging in (+)ssRNA viruses. As the author concludes by himself, “The molecular mechanism that controls selective packaging in (+)ssRNA viruses is very complex and depends on a series of interactions and characteristics that are specific to each species”. With this review, the author tries to put together and summarize the known data of many viral systems, providing sufficient details but not too much! This review can be a potential bibliographical source for new people in the field, covering a large panel of articles, from the 70’s to the actual papers.

- Overall, the figures/tables are too small! The table in particular is difficult to read on paper! Maybe passing a 7lines/14columns table to a 14lines/7columns can help.

- please, put all the “et al.” within the text in italic!

- Be carefull with the names of chemicals. For example, line 275, the author writes “a series of unpaired Guanines…”. Guanine is the name for the free nucleobase, not the nucleotide in the RNA. Should be “a series of unpaired Guanosyl residues…”. Please check in the whole text!

- line 195. The text should be Sorger and co-workers

- line 196. The text should be Qu and Morris

Author Response

Reviewer #2

We incorporated all suggestions, including changing the structure of Table 1.